

# Geo-referencing bird-window collisions for targeted mitigation

R. Scott Winton[1,2], Natalia Ocampo-Peñuela[1,2] and Nicolette Cagle[2]

[1] Department of Environmental Systems Science, ETH Zurich, Zurich, Switzerland
[2] Nicholas School of the Environment, Duke University, Durham, NC, United States of America

## ABSTRACT

Bird collisions with windows are an important conservation concern. Efficient mitigation efforts should prioritize retrofitting sections of glass exhibiting the highest mortality of birds. Most collision studies, however, record location meta-data at a spatial scale too coarse (i.e., compass direction of facing façade) to be useful for large buildings with complex geometries. Through spatial analysis of three seasons of survey data at a large building at a university campus, we found that GPS data were able to identify collision hotspots while compass directions could not. To demonstrate the broad applicability and utility of this georeferencing approach, we identified collision hotspots at two additional urban areas in North America. The data for this latter exercise were collected via the citizen science database, iNaturalist, which we review for its potential to generate the georeferenced data necessary for directing building retrofits and mitigating a major source of anthropogenic bird mortality.

## INTRODUCTION

Hundreds of millions of annual bird-window collisions collectively represent a major source of anthropogenic bird mortality in North America, second only to predation by feral cats (*Loss et al., 2014*). Researchers have hypothesized two principle perception hazards that glass poses for birds: transparency and reflectivity (*Snyder, 1946*; *Gelb & Delacretaz, 2009*). Transparency is problematic when parallel windows (such as in a glass tunnel) allow birds to see completely through a solid structure, especially if vegetation is visible on the other side. Reflected vegetation can trick birds searching for food or cover into colliding with a mirage. The severity of the conservation concern associated with bird-window collisions has motivated recent efforts to better understand the phenomenon in order to invoke mitigation strategies (*Gelb & Delacretaz, 2009*; *Bayne, Scobie & Rawson-Clark, 2012*; *Cusa, Jackson & Mesure, 2015*; *Ocampo-Peñuela et al., 2016*; *Sabo et al., 2016*), which could include installing bird-deterrent films on old windows (retrofitting) or bird-friendly architectural design. In the absence of any binding regulation for bird-friendly construction, mitigation is almost exclusively carried out on a voluntary basis by individual property owners or managers as an act of environmental charity or public relations (i.e., "greening").

Corresponding author
R. Scott Winton,
robert.winton@usys.ethz.ch,
scott.winton@gmail.com

Few studies have quantified the effectiveness of mitigation in practice (*Johnson & Hudson, 1976*; *Klem, 1990*; *Klem & Saenger, 2013*; *Ocampo-Peñuela, Peñuela Recio & Ocampo-Duran, 2015*; *Kahle, Flannery & Dumbacher, 2016*) and even in cases where collision avoidance can be estimated, the intangible value of saving bird lives is often outweighed by prohibitive costs. A recent high-profile example is the US Bank Stadium in Minneapolis, Minnesota, which has been covered by international news outlets. According to the media narrative, the designers of the Stadium ignored the warnings of conservation groups and built their stadium without bird-friendly design. In the migratory season following construction, surveys conducted by three local bird conservation organizations documented 74 window collisions, and the study authors conservatively projected that 360 birds would be killed in the three years post construction (*Audubon Chapter of Minneapolis, Minnesota Citizens for the Protection of Migratory Birds, Friends of Roberts Bird Sanctuary, 2017*). The chairwoman of the Minnesota Sports Facilities Authority justified inaction based on the cost of using bird-friendly glass, which she estimated to be US $60 million (http://www.mprnews.org/story/2015/01/16/bird-safe-glass-vikings). Indeed, if her estimate is accurate, the value of saving one bird life would have to be roughly US $17,000 for the investment to make financial sense over a 30-year horizon.

Resources available for conservation are always limited, and therefore the efficacy of bird-window collision mitigation efforts could be greatly improved by having a framework for setting priorities. In other words, it would be of great value to know which of the multitudes of windows in any given study are killing the most birds, so that the window panes exhibiting the highest levels of mortality could be retrofitted first. There are two fundamental obstacles to generating a list of high priority windows: (1) most studies of bird-window collisions do not routinely contribute data to a public access database, so data are not always available for analysis; and (2) the scale of location information is too coarse (i.e., compass direction of a building face) in typical multi-building survey protocols (*Borden & Lockhart, 2010*; *Hager et al., 2013*; *Kahle, Flannery & Dumbacher, 2016*; *Ocampo-Peñuela et al., 2016*).

A recent perspective paper by *Loss et al. (2015)* outlines an approach for aggregating existing local bird-collision survey data to help understand large-scale drivers of bird-window collisions. Critical omissions from the Google form data sheet (http://tinyurl.com/m3bvxel) suggested by *Loss et al. (2015)* are GPS coordinates, which are readily available to any citizen scientist with a smartphone. Such high precision spatial information could be valuable to conservationists, building managers and residents who clamor for practical and efficient retrofit solutions. Furthermore, the data platforms that *Loss et al. (2015)* recommend (Knowledge Network for Biocomplexity, http://knb.ecoinformatics.org/; the Global Biodiversity Information Facility, http://www.gbif.org/; http://www.Citsci.org) continue to lack any public data sets on bird-window collisions (as of October 2017), which indicates that scientists documenting bird collisions with windows have been slow to utilize these repositories.

In this paper, we investigate the website and companion mobile app, iNaturalist, as a potential platform for crowdsourcing, storing and analyzing bird-window collisions data. We present a case-study from Duke University in North Carolina, USA in which

we re-evaluate existing methodology by using iNaturalist to record GPS coordinates of individual collisions. Further, we examine crowd-sourced citizen-science collision data at two other urban areas in North America to test the potential of iNaturalist to identify both specific buildings and particular sections of façades that contribute disproportionately to bird deaths. We discuss the advantages and limitations of using iNaturalist versus other databases for storing records of bird-window collisions, with a focus on practical information for local conservation action at the scale of individual buildings.

## METHODS

### iNaturalist

iNaturalist is a citizen science database of natural history observations managed by the California Academy of Science (http://www.iNaturalist.org). Although it is not designed specifically for birds or even window collisions, the ''project'' feature of the database allows users to organize observations around a specific goal or theme. Species identifications are crowd-sourced voluntarily by the user-base and must meet an agreement threshold before observations are flagged as ''research grade''. Data can be entered through the website or using the mobile app for iPhone and Android. By using the mobile app, photos of organisms (e.g., bird-window collision victims) are automatically time-stamped and georeferenced using the smartphone's internal GPS, such that observations can be easily mapped. If data are entered using the website, the observer can place a georeferenced point either manually on a map, or by inputting GPS coordinates.

### Carcass surveys and study areas

We conducted six three-week bird-window collision surveys at six buildings on the campus of Duke University in Durham, North Carolina during spring and autumn migration seasons from 2014 through 2016. We followed methods described by *Hager et al. (2013)*, which have been adopted by dozens of university campuses across North America (*Hager et al., 2017*). We conducted surveys in the afternoon (roughly 14:00–16:00) and stored salvageable carcasses temporarily in a freezer before depositing them at the Museum of Natural History in Raleigh. For the three most recent surveys, we augmented the methodology (desbribed in more detail in *Wittig et al., 2017*) to report data to iNaturalist using the smartphone app, as a convenient way to record a GPS location and set of photos for each collision. We focus our analysis on one of the six buildings, the Fitzpatrick Center, which was responsible for the highest number of collisions in all six surveys. In response to student demands and media attention, Duke University installed a bird-deterrent grid of dots (Feather Friendly, Toronto, Ontario) on parts of the Fitzpatrick center façade shortly before the autumn 2016 survey (*Ocampo-Peñuela et al., 2016*). We compare collision data post-retrofit to three migration surveys, which predated the retrofit and lacked GPS location information, in order to evaluate the effectiveness of the retrofit and potential utility of our methodological augmentation. Since these six surveys in aggregate represent a very small data set without the benefits of experimental design (i.e., randomization, control groups, etc.), conventional statistical tests are not appropriate, so we limit our analysis and conclusions to qualitative statements.

In order to aggregate and track observations of window collisions at broader geographic scales, we started an iNaturalist project, "Bird Window Collisions" in April 2014. The project has since received more than 1,700 observations of 157 species from more than 160 unique contributors (as of October 2017). Two urban areas in the United States that have been particularly well-surveyed by local citizen science projects and lend themselves to a mapping and priority-setting exercise are downtown Baltimore, Maryland and the campus of The University of Pennsylvania (Philadelphia, Pennsylvania). At both cities, we selected a large building with complex geometry and high collision concentration to serve as an example for identifying hotspots. These buildings are the Baltimore Convention Center and at The University of Pennsylvania, a complex of four conjoined buildings connected by glass tunnels: Ryan Veterinary Hospital-Veterinary Medicine Old Quad-Rosenthal-Hill Pavilion.

## Mapping and identifying hotspots

To cull observations from the global bird-window collisions data set on iNaturalist not relevant to the above study sites, we sorted all observations by latitude. For each study site, we mapped all collisions reported and created a kernel density function, using the default Quartic (biweight) kernel, to highlight areas where collisions were concentrated using ArcMap 10.4.1. We then snapped collisions to the nearest point on the chosen building's perimeter if it was within 20 m (so as to exclude points recorded with low GPS precision) and used a point density function to highlight sections of perimeter building with the most documented collisions. Using both GPS and the conventional cardinal direction approach allowed us to contrast the relative effectiveness of differing location information at the building scale at the Fitzpatrick Center at Duke University. In order to highlight the benefits of using GPS coordinates over cardinal directions, we labelled collisions with the cardinal direction of the associated façade as reported by the observer. As cardinal directions where not reported as part of the Baltimore and University of Pennsylvania data collection efforts, we were unable to repeat this analysis at those sites.

## RESULTS

### Cardinal directions versus GPS points

We found cardinal direction data associated with collision events to be problematic, as exemplified by a large building with complex geometry, the Fitzpatrick Center at Duke University (Fig. 1A). When we mapped collisions coded by reported cardinal direction we found that this produced much ambiguity (e.g., a northwest façade being reported as either "N" or "W") (Fig. 1B). Even without the problem of ambiguous building faces, a simple tally of reported collisions for each cardinal direction (Fig. 1C) was of little use for local mitigation action. In contrast, applying a kernel density function to collisions mapped using iNaturalist (Fig. 1D) identified problematic sections of building perimeter that should be prioritized in mitigation efforts.

Data sets of 1,049 and 150 observations (as of March 2017) were submitted for downtown Baltimore and the University of Pennsylvania, respectively (Figs. 2A and 2B). These observations, which include GPS location metadata, lend themselves to identifying hotspots

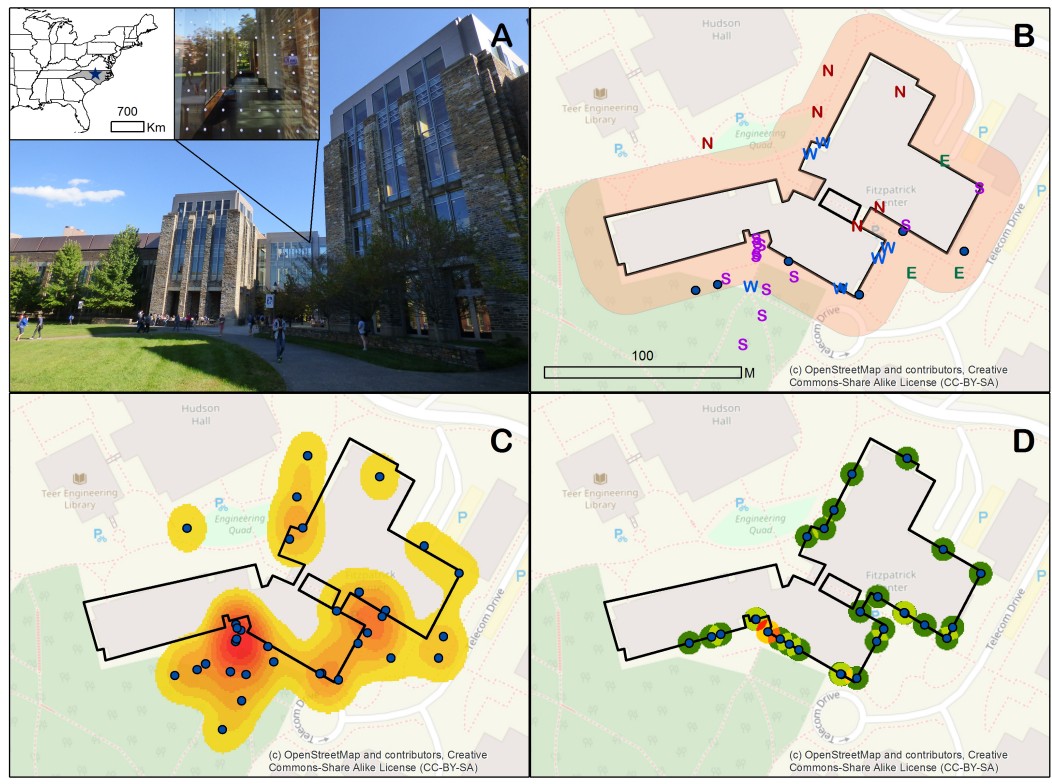

**Figure 1  Understanding bird-window collisions at the Fitzpatrick Center building at Duke University, North Carolina USA.** (A) Insets show the location of Duke University, and bird-friendly pattern of dots applied to some windows. (B) Reported cardinal directions of impacted building faces (with a 20-m buffer displayed); dots represent observations for which no data or letters other than N, E, S or W were entered. (C) A kernel density function of mapped collision points. (D) Collision observations snapped to the building perimeter and treated with a point density function.

at the façade scale, providing more precise information than is conventionally recorded by bird-window collision studies. By snapping collision locations to the borders of a four-building complex at the University of Pennsylvania (Fig. 2C) and to the Baltimore Convention Center (Fig. 2D) and applying a point density function highlights specific window sections that contribute disproportionately to collisions.

## Effectiveness of a retrofit at Duke University

The retrofit at Duke University's Fitzpatrick Center targeted architectural components that allow double-transparency (i.e., the potential to see sky or vegetation on the opposite side of a building through at least two windows on distinct building faces), including a glass tunnel and four towers that emerge above the roof line. These features were coated in a grid of opaque white dots spaced 5 × 5 cm apart (Feather Friendly; Toronto, Ontario) in September 2015. This mitigation strategy was based on our anecdotal observation of many collisions occurring at the glass tunnel (such tunnel collisions were not explicitly tracked in our original protocol). This mitigation effort, for which neither GPS points nor kernel density mapping information was available, essentially ignored the threat of

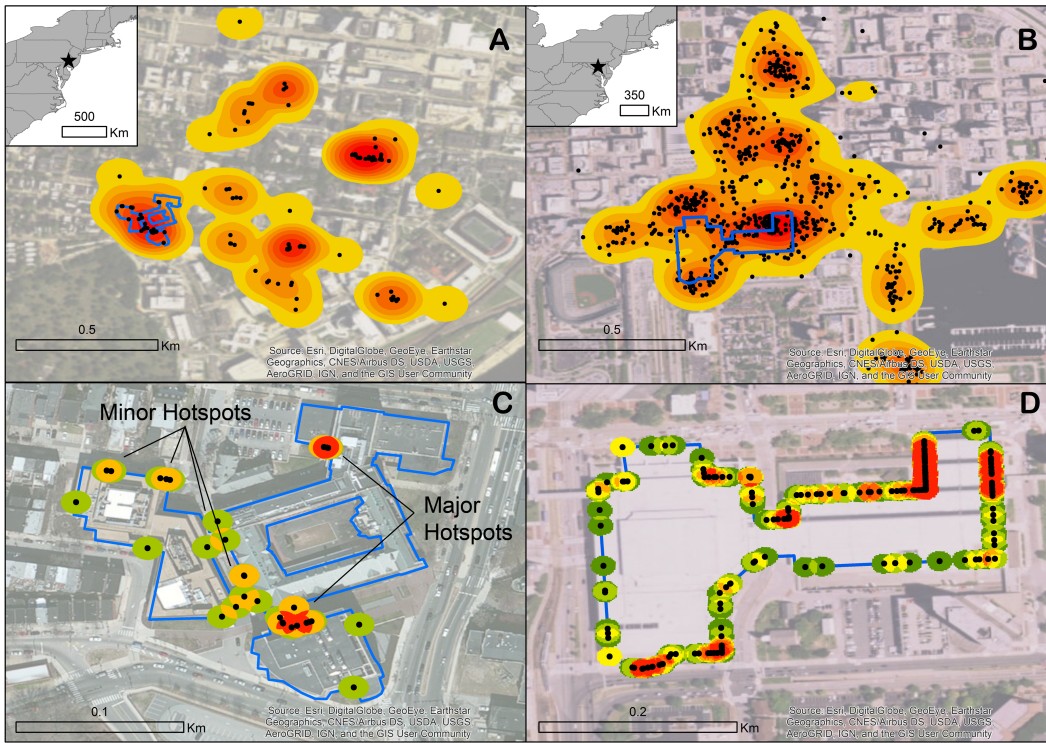

**Figure 2** Two examples of urban areas in the United States where extensive bird-window collision survey data reported to iNaturalist allow for the identification of hotspots via kernel density. University of Pennsylvania in Philadelphia (A) and downtown Baltimore (B) with perimeters of focal buildings outlined in blue. Collision points snapped to the perimeters of the Ryan Veterinary Hospital-Veterinary Medicine Old Quad-Rosenthal-Hill Pavilion complex (C) and the Baltimore Convention Center (D) with applied point density function to highlight most collision-prone segments of façade.

window-reflected foliage, which is far more difficult to target because bird perspectives and trajectories prior to collision are not known and reflections depend on the variable angle of the sun. In total, 437 m$^2$ of façade (minus some film wastage) were treated, representing less than one-quarter of the Fitzpatrick's total glass area.

Prior to the retrofit 31, 36 and 18 collisions were observed at the Fitzpatrick Center, representing 76%, 74% and 64% of total collisions detected during campus migration surveys from spring 2014 through spring 2015. Post-retrofit surveys detected 18, 3 and 16 collisions representing 67%, 60% and 47% of total collisions in autumn 2015 through autumn 2016. Although the post-retrofit collision counts and percentages are lower on average than those prior, it is statistically impractical to draw confident conclusions from such a small data set. Other windows are clearly still causing persistent collisions and the Fitzpatrick Center remains the deadliest building on Duke University's campus for birds. Nevertheless we found that the retrofit did successfully eliminate collisions at the glass tunnel demonstrating that retrofits are effective at preventing collisions at the windows they target.

## DISCUSSION

The case studies we have presented illustrate the utility of iNaturalist as a tool to track georeferenced bird-window collisions and we believe that widespread use could herald a new era of efficient bird-window collision mitigation. We provide the following critical review of the platform's features.

**iNaturalist is easy to use**. Among its many useful features, the smartphone app, which automatically attaches time and GPS location metadata to each collision observation, is especially valuable. This functionality eliminates tedious and time-consuming data entry steps and makes the platform highly user-friendly for reporting bird-window collisions. The convenience of contributing to iNaturalist via the phone app is probably the main reason why it has already received many hundreds of bird-window collisions observations. Other citizen science data repositories have added smartphone apps to increase data volume, such as eBird.org, which started in 2004 with a computer-based data entry portal and from January to March 2017 received 45% of its observations from mobile devices via its app launched in 2014 (eBird Project Coordinator, I Davies, pers. comm., 2017).

**iNaturalist data are easy to manage**. One major benefit of using iNaturalist is that it, like other open Citizen Science projects, empowers interested contributors to volunteer large amounts of data, relieving professional scientists of a major data collection burden. The iNaturalist platform also provides a crowd-sourced system for species identification, which greatly reduces the need for professionals to verify identifications. Because photos are embedded with each observation, spot-checking identifications for quality control is quick and easy. Observation locations are automatically mapped in the system's graphical user interface and data can be easily exported in spreadsheet format for subsequent analysis. No other platform provides this combination of convenient embedded functionality.

**iNaturalist is flexible**. Additional custom data fields can be added to provide metadata, such as whether the observation was part of a standardized survey and associated survey effort. This information is valuable for bridging the gap between localized window-collision surveys and generalizable scientific understanding of susceptibility (*Loss et al., 2015*). We did not originally add such fields when we started our project, and thus most of the data entered thus far must be treated as incidental even though the majority of entries are a part of formal surveys. This shortcoming can be rectified both prescriptively as observations continue to be added, as well as retroactively to previously recorded survey observations.

**iNaturalist is open-access**. This app is a free download and the database is open-access, meaning that anyone can report observations and anyone can download data for a species, an area, or a building in the case of bird-window collisions. Although downloading the data and mapping them in a specialized mapping program, such as ArcMap or QGIS is ideal for identifying mitigation hotspots (Fig. 2), managers can also visualize the data collected on the website using its built-in mapping feature, which can be useful for identifying areas with a high number of collisions without any knowledge of advanced mapping software.

**iNaturalist limitations**. As any data-collection platform, iNaturalist has its shortcomings. The main one that we have identified is the variable GPS precision. Under ideal conditions smartphone GPS precision is generally believed to be 4.9 m

(http://www.gps.gov/systems/gps/performance/accuracy), but it can vary depending on the openness of the area, the weather, and idiosyncrasies of smartphone and user behavior. During the study period the iNaturalist app was updated to display GPS "accuracy" (actually precision), which allows the user to wait for the GPS to stabilize at maximum precision before recording the point. In future studies, project leaders should predefine a target accuracy threshold that users should aim to achieve. Even with unconstrained scatter, we found the smartphone geolocations sufficient for prioritizing short sections of façade in our case studies. Future studies that constrain scattering of points to within 5–10 m will be able to more precisely target problematic windows and identify collision hotspots with lower volumes of data.

**iNaturalist alternatives**. To our knowledge only one other global-scale database used for tracking bird window collisions exists, the Fatal Light Awareness Project's FLAP Mapper (http://www.mapper.flap.org). Unlike iNaturalist, FLAP Mapper was developed specifically for tracking bird window collisions. It offers a lot of the same functionality as iNaturalist, such as photo imbedding and mapping and has received a large volume of observations (several thousand as of October 2017), especially in Canadian cities. Unlike iNaturalist, however, FLAP Mapper lacks a companion smartphone app, so users must manually navigate via the map interface on a web browser to place a data point where a carcass was found. Not only does the lack of an app make for a less streamlined user-experience, but it also leads to data of unknowable spatial accuracy and precision. Do users place their points with enough precision to identify specific windows or façades? The instructions (http://www.flap.org/mapper_guide.php; accessed March 2017) merely say "zoom in as you see fit". A GPS point from a smartphone can also be imprecise, but iNaturalist also provides metadata quantifying this source of error, whereas human input error is much more difficult to handle in an objective way.

## CONCLUSIONS

After decades of bird-window collision research, examples of successful applications of bird-friendly construction and retrofits are finally starting to accelerate thanks to leadership led by scientists and organizations such as National Audubon Society and American Bird Conservancy. Our analysis suggests that iNaturalist may be a tool that has the potential to increase the volume of data collection through its app and also facilitate improvements in the cost-efficiency of retrofitting efforts through its geospatial metadata.

We invite existing window-collision survey efforts, as well as newly developed programs, to embrace iNaturalist. As we demonstrate for Duke University and two other urban sites, mitigation actions can be prioritized to select windows on relatively small sections of façade, which could make retrofits cheaper and therefore far more likely to be implemented.

## ACKNOWLEDGEMENTS

We thank the many Duke University graduate and undergraduate students who assisted with data collection for this project, especially the Wildlife Surveys and Biodiversity Issues and Field Methods classes at the Nicholas School of the Environment. We also thank the

many volunteers around campus who have collected data for the collision project since 2013, especially Anna Wilson. John Gerwin of the North Carolina Museum of Natural Sciences assisted by receiving all bird carcasses we collected. We thank Alfonso Alonzo and the Duke University Facilities and Management Department for their willingness to take action on behalf of campus birds by retrofitting Fitzpatrick. We also acknowledge the contributions to the Bird-window Collisions iNaturalist project made by 164 users, but especially Aaron Heinsman and others of Lights Out Baltimore and Joe Durrance of the University of Pennsylvania.

### Funding

During the time this research was carried out, Robert Scott Winton was funded by an Anne T and Robert M. Bass fellowship; and Natalia Ocampo-Peñuela was funded by a Colciencias-Fulbright scholarship. The funders had no role in study design, data collection and analysis, decision to publish, or preparation of the manuscript.

### Grant Disclosures

The following grant information was disclosed by the authors:
Anne T and Robert M. Bass fellowship.
Colciencias-Fulbright scholarship.

### Competing Interests

The authors declare there are no competing interests.

### Author Contributions

- R. Scott Winton conceived and designed the experiments, performed the experiments, analyzed the data, wrote the paper, reviewed drafts of the paper.
- Natalia Ocampo-Peñuela conceived and designed the experiments, performed the experiments, analyzed the data, contributed reagents/materials/analysis tools, prepared figures and/or tables, reviewed drafts of the paper.
- Nicolette Cagle conceived and designed the experiments, performed the experiments, reviewed drafts of the paper.

### Data Availability

iNaturalist: https://www.inaturalist.org/projects/bird-window-collisions.

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
