# Peer review of "Geo-referencing bird-window collisions for targeted mitigation"

_PeerJ, doi:10.7717/peerj.4215_

## Round 0.1 · original submission · Minor Revisions

Please make your revisions in light of the comments/suggestions from reviewer 2 and reviewer 3.

·

Basic reporting

Clear, concise, an accurate professional English used throughout. Except for Nelson (2015) which I may have missed in the text, all other citations are appropriately in text and the Reference section. The figures and supplemental materials are useful, appropriate, and complement the text. Meets all your standards in my view.

Experimental design

Although a rigorous experimental design is not appropriate given the objectives of this study, the authors adequately explain this in detail and provide extensive justification for what was done, how, and the marked value for future studies; fundamentally, providing a means of gathering more efficient and comprehensive quantitative data on incidents of these important conservation events. Ample information and description is provided for replicating and enacting the proposed advancements. Meets all your standards in my view.

Validity of the findings

Introduces and encourages an enhancing data gathering technique to aid current and future work on this subject. Conclusions are generally meaningful, useful, and supported by case study results presented and interpreted. A valuable contribution is the comparison to other online sources (programs - eBird, FLAP Mapper, but why not dBird (?) from New York City Audubon) with similar intents. The authors offer detailed and convincing evidence that their recommended iNaturalist use (methods of data entry) offers the most utility. Meets all your standards in my view.

Additional comments

Compliments for recommending a useful and valuable tool to further study this important conservation issue for birds and people.

Reviewer 2 ·

Basic reporting

I feel the manuscript could be tightened up in places, though my criticisms are all quite minor. They are detailed here.

Presentation of information:

Introduction -

Line 20: I would mention the leading cause of anthropogenic bird mortality and how bird-window collisions compare in terms of number of casualties.

Lines 27-34: I suggest combining the first three paragraphs into two, with the break after “(i.e. ‘greening’)”. The current paragraph breaks interrupt the narrative.

Lines 34-50: This is useful as a case study but the paragraph overall is difficult to read due to the many internal citations. How stable are the urls cited here? Are they all necessary? Consider paring down this paragraph.

Lines 61-64: This sentence (“The goal of….”) is awkward in context; needs smoothing to better align with previous and following sentences


Methods -

Lines 99-101: You should mention explicitly what led to Duke's decision to retrofit the building.

One glaring omission from the Methods is the survey methodology itself. When were surveys done? What time of day? What was done with the carcasses - were they salvaged and deposited in a museum? Was any information noted on weather conditions, wind direction, amount/height of vegetation surrounding windows, etc? How exactly was iNaturalist used on the ground during your surveys?


Results -

Lines 155-161: I think this background - specifically, why birds collide with windows, including references dating back to 1946 - should be raised earlier in the paper. By doing this, the summary of the effectiveness of the retrofit would be more succinct.

Lines 163-165: Reference these percentages to survey years - 2014-2016.

Line 168 - this is important to highlight, that retrofitting can successfully eliminate collisions! I would add emphasis and broader context to this point.


Minor copy edits:

Line 13 - throughout the paper, you use data as a singular noun. While technically correct either way, I think your sentences read better when "data" is treated as plural.

Line 39 - the franchise has been, not have been

Line 44 - insert comma after "collisions"; insert "the" after "and"

Line 51 - insert comma after "limited"

Line 57 - see comment on "data" above

Line 66 - smartphone should be one word

Line 110 - "that have been"

Line 112 - cities, not areas

Line 115 - "four-conjoined" should not be hyphenated

Line 124 - delete "building"

Line 132 - delete "is"

Line 163 - insert comma after "Center"

Line 172 - insert comma after "collisions"

Line 177 - delete "a"

Line 184 - see comment on "data" above

Experimental design

No comment (see "Validity of the findings" below)

Validity of the findings

I think this is a great paper that furthers the cause of citizen science initiatives by introducing iNaturalist as a tool to provide useful and replicable data in a specific context. Although the design of the present study is beset by problems, including small sample size and lack of controls, the authors are well aware of this and state explicitly that their conclusions should be interpreted with this in mind. To me, the chief contribution of this paper is to highlight the rigor that iNaturalist provides over subjective descriptions of survey locations provided by study participants. This is something that is badly needed in many citizen science endeavors, and widespread adoption of the methodology presented here will improve the quality and replicability of all studies for which georeferenced localities are an essential component of the dataset. The study therefore has applicability far beyond documentation of bird-window collisions, and this broader benefit to the scientific community far outweighs any shortcomings in their survey design. I would add that, based on my own experiences monitoring bird-window collisions, I am quite confident that greater sample sizes would only bolster their conclusions, and that appropriate controls would be difficult or impossible to incorporate into any study on bird-window collisions, considering the practical and legal issues involved with surveying buildings. The authors have done a commendable job within these constraints.

Reviewer 3 ·

Basic reporting

The writing is clear and understandable with only minor edits needed, see below. The literature cited is satisfactory but does include several incomplete references.

Experimental design

The authors point out that there is insufficient experimental setup to warrant statistical analyses. The reporting is mostly qualitative in nature but is sufficient given the nature of the approach.

Validity of the findings

The authors have done a good job of summarizing the data.

Additional comments

Manuscript 21227
PeerJ
Overall Assessment
The authors report on a new method to better track the location of birds killed by window strikes. Most studies simply record the wall of the building, or perhaps the direction of the wall, at a very coarse scale. Without more specific information, attempts at mitigation may be more difficult if there are particular areas of the wall that are more likely to be prone to bird strike. The authors argue that GPS locations are better than simply compass directions. The authors apply these findings to two other locations where data was input using iNaturalist, a citizen science program for reporting natural history observations. The authors recommend using iNaturalist in an effort to thwart large-scale bird-window collisions.
The authors point out that a recent approach used by Loss et al (2015) involves a google form datasheet that lacks a space for GPS coordinates. In addition, the data platforms recommended by Loss et al do not contain any public datasets so the present authors warn that these databases are not being used by scientists or members of the public interested in bird-window kill data. As a result, the authors recommend yet another platform, iNaturalist, because it allows for GPS coordinates, crowd-sourced identification, and citizen science data collection.
Because the dataset is small and without proper control design, the authors do not use statistics but, instead, report the results as qualitative statements.
I think figure 1 illustrates quite well what the issues are relative to using simple cardinal direction vs GPS coordinates. The kernel function in figure 1C and the point density function in 1D show clearly where the problem areas on the buildings are.
Overall I think this approach is important and will make a nice contribution to the burgeoning field of bird-window mortalities.
Line Edits
20 If you are going to tease the reader with “second-greatest”, you should probably let them know what the highest source is.
25 “bird-deterrent films”, compound modifier should be hyphenated
29 Inconsistent single vs double quotes for quotations. See, for example, line 227 where the authors write “zoom in as you see fit”
39 franchise is singular so it should be “has been publicly”
40 The authors imply that it was up to the Minnesota Vikings to invest in a bird-friendly facility. But, the stadium is not owned by them. “U.S. Bank Stadium is owned by the state of Minnesota, represented by the Minnesota Sports Facilities Authority. The Minnesota Sports Facilities Authority (MSFA) was established by the legislature in 2012 and charged with the design, construction and operation of U.S. Bank Stadium. The MSFA consists of five members and is chaired by Michele Kelm-Helgen. The other members are John D. Griffith, Tony Sertich, Bill McCarthy and Barbara Butts Williams. Ted Mondale serves at the CEO/Executive Director.” Source: https://www.usbankstadium.com/stadium-info/faq. As such, it seems the criticism should be redirected to the state of Minnesota, not to the Vikings.
42 The authors should reference the statement “according to the popular narrative…”
57 Data are plural so it should read “so data are not always”
65 coordinates are plural so it should be “suggested by Loss et al. (2015) are GPS coordinates”
94 Better to use “autumn” instead of “fall”. The latter is generally only used by North Americans.
96 Statement about methods being adopted by dozens of campuses should be referenced.
110 Suggest inserting “that” so it reads “Two urban areas in the United States that have been” to match the syntax “are downtown….”.
129 The abbreviation “Penn” has not been formally introduced.
136 Better to write “collisions reported”
233 (and 45) Better to use the full name National Audubon Society. “Audubon” is short form.
Reference
Inconsistent capitalization for titles. Bayne et al uses Sentence style. But, Borden and Lockhart capitalize each word.
Audubon reference is incomplete.
Forbes reference is incomplete.
Klem 1990 does not list the journal name.
Nelson reference is incomplete.
Ocampo-Peñuela et al. 2016 does not list journal name.

---

## Round 0.2 · accepted · Accept

The manuscript has been revised satisfactorily in light of the reviewers' comments.